# The Role of Water Content in the Casing Layer for Mushroom Crop Production and the Occurrence of Fungal Diseases

**María J. Navarro [1],\*** , **Jaime Carrasco [2,3],\*** and **Francisco J. Gea [1]**

1   Centro de Investigación, Experimentacion y Servicios del Champiñón (CIES),
    16220 Quintanar del Rey, Cuenca, Spain; fjgea.cies@dipucuenca.es
2   Centro Tecnológico de Investigación del Champiñón de La Rioja (CTICH), 26560 Autol, la Rioja, Spain
3   Department of Plant Sciences, University of Oxford, South Parks Road, Oxford OX1, UK
\*   Correspondence: mjnavarro.cies@dipucuenca.es (M.J.N.); jaime.carrasco@plants.ox.ac.uk (J.C.);
    Tel.: +34-967-496-198 (M.J.N.); +34-941-390-960 (J.C.)

**Abstract:** Mushroom cultivation requires effective control of environmental cues to obtain the best yield and high quality. The impact of water content in the casing layer on mushroom yield and the incidence of two of the most important diseases in the mushroom growing farms, dry bubble and cobweb diseases, was evaluated. Different initial water content in the casing and two alternative irrigation programs applied (light or moderate irrigation) were the agronomic parameters under study during five separate button mushroom crop trials. Higher initial humidity content in the casing layer reported a larger yield, with a fewer number of basidiomes but heavier, while no correlation to the dry matter content or the colour of the basidiomes was noted. The incidence of dry bubble disease was not conditioned by the water content of the casing layer, at the high moisture levels established in the study. In the case of *Cladobotryum mycophilum*, the lower moisture level of the casing layer reported more incidence of cobweb disease, and subsequently harmful yield losses. According to the results obtained, the right management of the moisture level in the casing materials could promote crop yield and preclude the significant impact of dry bubble and cobweb diseases.

**Keywords:** peat; water content; *Agaricus bisporus*; *Cladobotryum mycophilum*; *Lecanicillium fungicola*





## 1. Introduction

The commercial cultivation of button mushrooms (*Agaricus bisporus* (Lange) Imbach) is an intensive horticultural process that requires two different substrates: a selective compost, which is a nutritive reservoir for mycelium growth and development, and the casing layer, that is placed on top of the colonized compost and built up with specific materials to induce and favor the development of basidiomes [1,2]. Correct water content in the casing is required to provide the essential moist microclimate that supplies water for the growth and development of the crop while facilitating the transport of dissolved nutrients and prevents compost surface from drying out [1].

The composting process to achieve the selective substrates passes through several mesophilic and thermophilic stages along with a succession of bacterial and fungal populations to modify the raw materials [3]. The compost is pasteurized during the phase II process to eliminate parasites and competitors for the host before spawning [4]. The casing material also presents a diverse fungal and bacterial microbiome, whose interaction with the host mycelium is diverse and not well described [5–7]. Some of these casing inhabitants are required for the growth and fructification of cultivated mushrooms [8]. Casing composition and the casing microbiota may also affect the development of mushroom crop diseases [9], among the most damaging of which are dry bubble disease caused by *Lecanicillium fungicola* (Preuss) Zare and W. Gams, and cobweb disease caused by *Cladobotryum mycophilum* (Oudem.) W. Gams and Hoozem [10–12].

Mushroom growers are currently applying a casing material based on black peat, although environmental concerns advise against this choice (the decreasing availability and the environmental impact on peatlands predict increasingly restrictive legislation for peat mining) [13]. Basiome obtains water mainly from the casing layer, depending on the availability of water in the compost and the casing; casing material acts as a water reservoir essential for mycelium growth and basidiome development while protecting compost from drying [14]. Depending on the crop necessities and the precocity of the harvest desired, growers must decide when to water, how much water is needed and how to supply the resource. It is important to remark that an appropriate balance between water supply and demand (including evaporation as moisture sink) must be achieved for correct cropping and also for disease control since persistent moisture in the casing or basidiomes is been related to the germination of pathogenic spores and the growth of harmful bacteria [15]. The irrigation of the casing layer (the amount of water supply, the irrigation pattern and the system of application) has a great influence on the success or failure of a mushroom crop [16–19]. Suboptimal water supply can result in desiccation of the casing, loss of the required structure and porosity in the material, and, as a result, reduction in yield and quality, and hampering of subsequent irrigations [20]. Over-watering while cropping can lead to greater susceptibility to mushroom diseases such as bacterial blotch [21,22], internal stipe necrosis [23,24] or cobweb disease [12], also driving to a reduction in yield and quality.

During the 2010s, an outbreak of cobweb disease was registered in Spanish mushroom crops, concurring with the transition to the currently used peat-based casing instead of the historically used mineral soil [12]. This important outbreak was related to the higher water holding capacity (WHC) of the peat, ultimately disposing of an environment with better conditions for the viability of the harmful conidia [9,25]. However, Happ and Wuest [26] had previously suggested that *L. fungicola* was more severe when the casing was too dry, but the observed symptom differed depending on its water content: under moisture stress, dry bubble disease of mushroom growing on peat moss usually occurred as the undifferentiated amorphous mass of mushroom tissue, whereas on dry mineral soil, cap spotting or necrotic lesions were most frequently observed; inoculating the disease at the time of pin formation (day 12 after casing) generally resulted in lower mushroom yield. The type and severity of disease symptoms are directly influenced by the moment of infection, and the evidence reported suggests that the *Agaricus* crop is most sensitive to the parasite before the formation of basidiomes [9].

The accurate control of agronomical conditions during cropping together with environment-friendly treatments are efficient measurements to design Integrated Disease Management (IDM) programs in mushroom crops and to overcome the dependence on chemical fungicides. In this sense, the standardization of the moisture content in the casing layer is key aspect of the agronomy of button mushrooms. This paper analyses the effect of different levels of moisture in the casing layer in relation to the incidence of two of the most important fungal diseases in the mushroom growing farms, dry bubble and cobweb disease. Moreover, agronomic valuation of crop performance under these setting conditions was also carried out to maximise crop yield. Ultimately, the results achieved contribute to understand the role of the water content in the casing to maximise mushroom yield and as a cultural input to prevent fungal infections.

## 2. Materials and Methods

### 2.1. Agronomic Evaluation of the Crop

By mid-2018 a cropping trial (Trial A) was set up in an experimental mushroom growing room. The trial was performed using 32 plots (0.15 m$^2$ in area) containing commercial phase III compost (Champinter SCL, Villamalea, Albacete, Spain) (characterisation of the composts used in the trials is summarised in Table S1) and spawned at 1% with the commercial mycelium (Laboratorio, Champinter SCL, Villamalea, Albacete, Spain). On casing day (day 0) of the cropping cycle, the compost was cased with a peat-based casing as a 35–40 mm thick layer (5.5 L per experimental tray) (characterization of the casing materials used in the trials is summarised in Table S2). All the plots were cased with the

same casing material, differing only in the initial moisture rate. The amount of water in growing media has been expressed as a volumetric rate [27]. The experiment compared the effect of two factors, initial volumetric water content of casing and irrigation program. Trial A was set up as a baseline to evaluate the agronomic performance of the crop. Four treatments were defined in the trial, with eight plots per treatment: low initial water content of the casing with a light irrigation program (H1R1), high initial water content of the casing with a light irrigation program (H2R1), low initial water content of the casing with a moderate irrigation program (H1R2), and high initial water content of the casing with a moderate irrigation program (H2R2). The irrigation program is shown in Table 1. The volumetric water content (θ) of casing materials was monitored at regular intervals through the crop cycle using a GP1 Datalogger with SM-150T moisture sensor (Delta-T Devices Ltd., UK, Cambridge).

**Table 1.** Initial water content ($m^3 \, m^{-3}$) of the casing layer and irrigation program applied ($L \, m^{-2}$) in the trials.

| Trial | Initial Water Content ($m^3 m^{-3}$) | | Irrigation F1 * ($L \, m^{-2}$) | | Irrigation F2 ¥ ($L \, m^{-2}$) |
|---|---|---|---|---|---|
| | H1 | H2 | R1 | R2 | R1 and R2 |
| A | 0.66 | 0.71 | 3.3 | 3.3 + 1.4 | 2.8 |
| L-1 | 0.62 | 0.71 | 3.3 + 0.7 | 3.3 + 2.0 | - |
| L-2 | 0.69 | 0.74 | 3.3 + 0.7 | 3.3 + 2.0 | - |
| C-1 | 0.60 | 0.71 | 3.3 + 0.7 | 3.3 + 2.0 | 2.7 |
| C-2 | 0.55 | 0.71 | 3.3 + 0.7 | 3.3 + 2.0 | 3.7 |

* Days 23 and 24 after casing, end of the first flush (F1). ¥ Day 31 after casing, end of the second flush (F2). R1: low irrigation program; R2: moderate irrigation program.

The main production and quality parameters (earliness, number of basidiomes, yield, colour, unitary weight and dry matter content) were evaluated. Basidiomes were harvested daily at their optimal commercial development stage, counting and weighting the basidiomes picked from each block. Yield is expressed as kg per $m^2$. Earliness, or days to the first harvest, was expressed as the number of days between the casing and the beginning of the harvest of the first flush. On the day when most basidiomes were picked during the three flushes, the dry matter and colour were recorded. Dry matter and water contents were determined by measuring weight loss after oven drying at 105 °C for 72 h (Memmert GmbH + Co.KG, Schwabach, Germany, Model UFE 600). Basidiome colour was determined using a Minolta CR-300 spectrophotometer (Konica Minolta, Tokyo, Japan), with 15 measures per flush and treatment.

### 2.2. Evaluation of the Disease Incidence

Four trials were set up in order to test the incidence of dry bubble and cobweb diseases (Table 1). Two separate trials were performed for each disease. Each trial consisted of two groups, control uninoculated plots and plots artificially inoculated with the pathogen. Four treatments within each group, as described in the previous section, were set up: H1R1, H2R1, H1R2 and H2R2. The irrigation programs are included in Table 1. Six replicates per treatment and group were evaluated. The cultural practices were the same described in trial A. The volumetric water content (θ) of casing materials was monitored at regular intervals through the crop cycle by GP1 Datalogger with SM-150T moisture sensors.

Trials with *Lecanicillium fungicola* were carried out from October to December 2018 (Trials L-1 and L-2), while the trials infected with *Cladobotruym mycophilum* were carried out from January to May 2018 (Trials C-I and C-2). For each trial, experimental plots (0.15 $m^2$) filled with 10 kg commercial phase III compost (Champinter SCL, Villamalea, Albacete, Spain) (Table S1) and spawned at 1% with the commercial mycelium Triple X (Amycel Spawn Mate, Ittervoort, The Netherlands) were used. On casing day (day 0) of the cropping cycle, the compost was cased with a peat-based casing of 35–40 mm thick layer (5.5 L per experimental tray). All the plots were cased with the same casing material (Table S2), differing only in the initial moisture content (Table 1). The plots were disposed of in shelves at three different levels

in a completely randomized block design. Twelve or ten days after casing (L and C trials, respectively), 24 blocks were inoculated with a conidial suspension ($7 \times 10^3$ conidia mL$^{-1}$) of *L. fungicola* strains LF18-1 (trial L-1) and LF18-3 (trial L-2), or the *C. mycophilum* strains LR16.3 (trial C-1) and MJ-1 (trial C-2). Each block was sprayed (20 mL per block) by pipetting onto the surface of the casing layer at a rate of $10^6$ conidia m$^{-2}$. The remaining blocks (24 blocks) were sprayed with sterile distilled water as a control. The basidiomes were harvested daily. The number and the total weight of fruiting bodies were recorded for each block.

In the case of *L. fungicola* trials, the basidiomes and dry bubbles were separately harvested daily during two flushes. Harvested basidiomes were classified as either healthy or infected by *L. fungicola*. Disease incidence was stated as the ratio of diseased basidiomes vs. the total number of harvested basidiomes (healthy and diseased) [28].

The effect of cobwebs on mushroom productivity was evaluated by comparing mushroom yield (kg m$^{-2}$), harvested during three flushes, and also based on cobweb patches detected on the casing material to quantify the area colonized by cobweb as previously detailed [12,29].

### 2.3. Statistical Analyses

ANOVA was used to analyze the data, after transformation (if necessary) to stabilize variances, and the Tukey-HSD test was employed for mean separation. Significance was evaluated at $p < 0.05$ in all tests. Statistical analyses were performed using Statgraphics Centurion XVII software (Statistical Graphics Corp., Princeton, NJ, USA).

### 3. Results

#### 3.1. Agronomic Evaluation of the Crop

The impact of initial water content in the casing and irrigation management on agronomic and quality parameters for the crop was evaluated (trial A). Figure 1 shows the volumetric water content in the casing layer during the different crop cycles for the treatments established on the study. The figure shows the volumetric water content in the casing along trial A.

The two initial values are 0.66 and 0.71 m$^3$m$^{-3}$ for H1 and H2, respectively. On day 24, after the harvesting of the first flush, the irrigation program distinguished four treatments (Table 1).

Yield (kg m$^{-2}$) harvested in Trial A (Table 2) was significantly the largest ($F_{3.31}$ = 12.04, $p$ = 0.0000) in the casing layer with the greatest humidity level (H2R2). The increase of the yield for H2R1 was up to 20% higher, comparing with H1R1 (23.5 vs. 19.4 kg m$^{-2}$, respectively). It is also important to underline that while a higher initial humidity in the casing layer reported a statistically significant higher production (H2R1 vs. H1R1; H2R2 vs. H1R2), the upper irrigation program also affected but just lightly increasing the yield (H1R2 vs. H1R1; H2R2 vs. H2R1). Furthermore, the largest percentage of big basidiomes (cup diameter > 40 mm, Table 2) was registered within the high water treatments, which facilitated the harvest. The earliness of the first flush did not show statistical differences between the treatments ($F_{3.31}$ = 0.53, $p$ = 0.4710), with values corresponding to 23.3 days after casing. Regarding the number of basidiomes harvested per square meter (Table 2), those blocks with the lowest level of moisture registered the highest value (H1R1), showing statistically significant differences with the treatments H2R1 and H2R2 ($F_{3.31}$ = 2.96, $p$ = 0.0496).

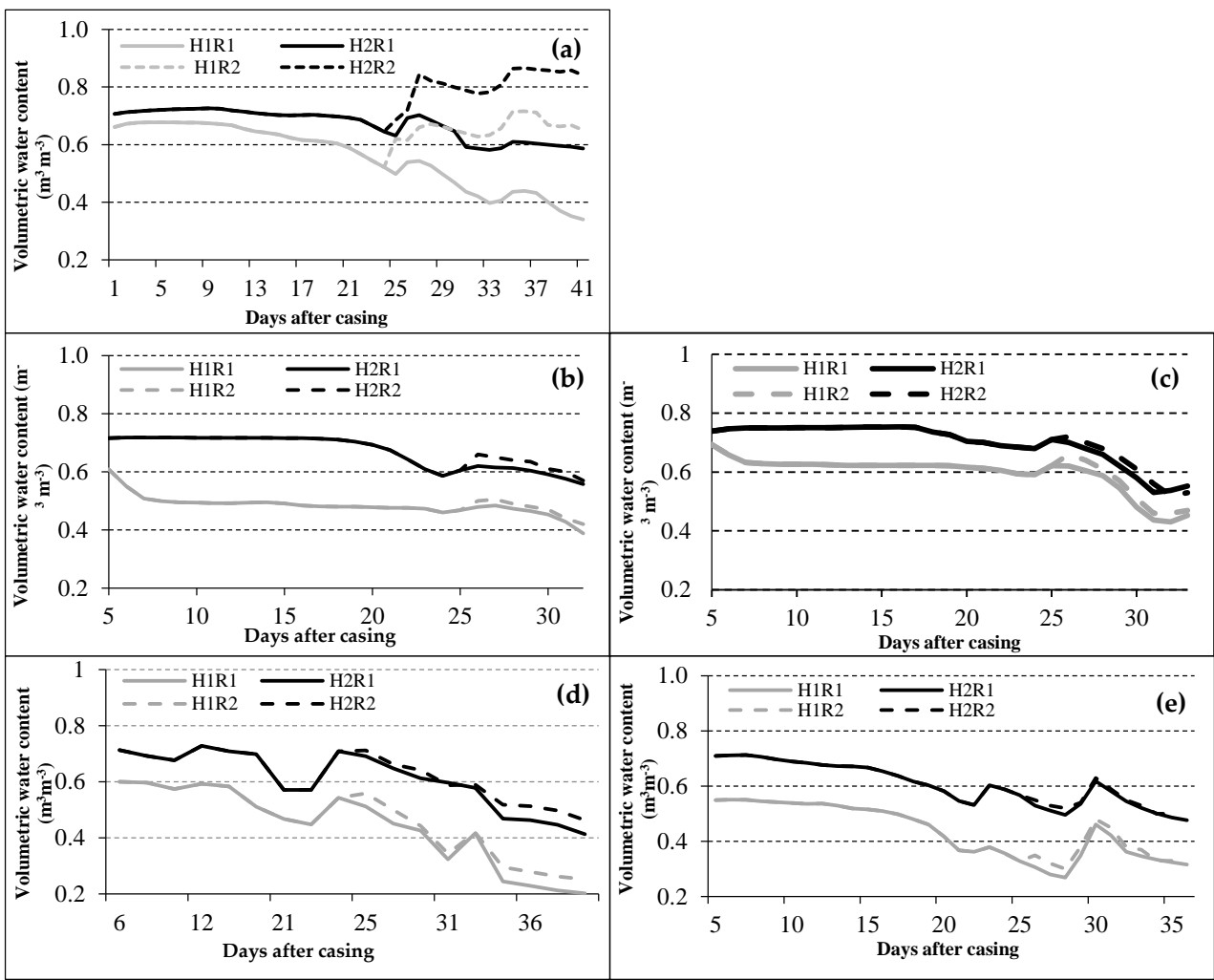

**Figure 1.** Volumetric water content in the casing for the treatments established in the trials. H1R1, low initial water content with a light irrigation program; H2R1, high initial water content with a light irrigation program; H1R2, low initial water content with a moderate irrigation program; H2R2, high initial water content with a moderate irrigation program. (**a**) trial A; (**b**) trial L-1; (**c**) trial L-2; (**d**) trial C-1; (**e**) trial C-2.

**Table 2.** Main quantitative and qualitative production parameters assessed in Trial A (yield, number of basidiomes and dry matter content).

| Treatment | Yield (kg m⁻²) | | | Basidiomes (n° m⁻²) | Dry Matter (%) | | |
|---|---|---|---|---|---|---|---|
| | Total | $\emptyset > 40mm$ | $\emptyset < 40mm$ | | 1st Flush | 2nd Flush | 3rd Flush |
| H1R1 | 19.4 ± 1.6 a | 12.70 ± 2.3 a | 6.7 ± 1.7 b | 136 ± 13.9 b | 9.8 ± 0.5 a | 9.0 ± 0.3 ab | 9.7 ± 1.0 a |
| H2R1 | 21.4 ± 1.0 b | 17.5 ± 2.2 b | 3.8 ± 2.1 a | 113.6 ± 25.0 a | 10.3 ± 0.7 a | 9.5 ± 0.9 b | 9.1 ± 0.9 a |
| H1R2 | 20.3 ±1.5 ab | 15.3 ± 1.3 ab | 5.0 ±1.3 ab | 126.5 ±15.3 ab | 10.3 ± 0.2 a | 8.5 ± 0.4 a | 9.7 ± 0.9 a |
| H2R2 | 23.5 ± 1.6 c | 20.7 ± 2.3 c | 2.9 ± 1.9 a | 114.1 ± 14.2 a | 10.4 ± 0.8 a | 8.9 ± 0.4 ab | 8.8 ± 0.5 a |
| $F_{3.31}$ | 12.04 | 20.92 | 7.08 | 2.96 | 0.71 | 2.14 | 1.20 |
| P | 0.0000 | 0.0000 | 0.0011 | 0.0496 | 0.5662 | 0.1480 | 0.3515 |
| SED | 0.510517 | 0.738002 | 0.623188 | 6.26 | 0.3 | 0.3 | 0.4 |
| LSD | 1.47891 | 2.85018 | 2.40676 | 18.13 | 0.9 | 0.9 | 1.3 |

Numbers within a column followed by a common lowercase letter do not differ significantly according to Tukey's HSD test at $p = 0.05$. F: F value; *p*: *p*-value; SED: Standard Error of the Difference; LSD: Least Significant Difference. H1R1, the low water content of the casing with a light irrigation program; H2R1, high water content of the casing with a light irrigation program; H1R2, low water content of the casing with a moderate irrigation program; H2R2, high water content e of the casing with a moderate irrigation program.

With respect to the quality parameters evaluated in the trial (dry matter and colour), statistical analysis did not show significant differences among treatments (Tables 2 and 3).

**Table 3.** Colour as qualitative parameter of the basidiomes harvested in the three flushes for the treatments set up in trial A.

| Treatment | 1st Flush | | | 2nd Flush | | | 3rd Flush | | |
|---|---|---|---|---|---|---|---|---|---|
| | L * | b * | AE * | L * | b * | AE * | L * | b * | AE * |
| H1R1 | 94.4 a [‡] | 9.0 a | 9.5 a | 94.4 a | 9.9 a | 10.3 b | 92.1 a | 10.7 a | 12.1 a |
| H2R1 | 94.7 a | 9.0 a | 9.4 a | 94.7 ab | 9.0 a | 9.4 ab | 93.5 a | 9.3 a | 10.2 a |
| H1R2 | 94.5 a | 8.9 a | 9.4 a | 95.12 b | 9.8 a | 10.0 ab | 92.9 a | 10.4 a | 11.3 a |
| H2R2 | 94.6 a | 8.8 a | 9.2 a | 94.8 ab | 8.9 a | 9.3 a | 92.0 a | 10.2 a | 12.1 a |
| $F_{3.15}$ | 0.39 | 0.33 | 0.43 | 2.60 | 2.6 | 2.58 | 0.40 | 1.62 | 0.96 |
| *p* | 0.7594 | 0.8033 | 0.7384 | 0.1002 | 0.1005 | 0.1018 | 0.7527 | 0.2402 | 0.4455 |
| SED | 0.21 | 0.17 | 0.19 | 0.19 | 0.30 | 0.30 | 1.17 | 0.45 | 0.88 |
| LSD | 0.66 | 0.53 | 0.57 | 0.59 | 0.94 | 0.94 | 3.61 | 1.40 | 2.97 |

[‡] Numbers within a column followed by a common lowercase letter do not differ significantly according to Tukey's HSD test at *p* = 0.05. H1R1, low water content of the casing with a light irrigation program; H2R1, high water content of the casing with a light irrigation program; H1R2, low water content of the casing with a moderate irrigation program; H2R2, high water content e of the casing with a moderate irrigation program.

*3.2. Evaluation of the Dry Bubble Disease Incidence*

The impact of initial water content in the casing and irrigation program on the incidence of dry bubble disease was evaluated in artificially inoculated crop trials while analysing disease impact on yield loss. The initial water content of the casing layers was 0.62 and 0.71 $m^3 m^{-3}$ in trial L-1, and 0.69 and 0.74 in trial L-2, for H1 and H2 treatment, respectively (Table 1). On day 24, after the harvesting of the first flush, the irrigation program distinguished the four treatments (Figure 1b,c for Trials L-1 and L-2, respectively).

In trial L-1, disease incidence values were around 30% at the first flush for all the treatments (24.1–35.6%), without statistical differences between them (Figure 2a). During the second flush, disease levels increased to 60% in all the treatments (60–79%), but statistical differences were neither detected between them. Likewise, in trial L-2, there were no statistical differences between treatments, although disease incidences were lower in absolute terms (<25% for any treatment and flush).

The yield loss associated to dry bubble disease (Figure 2b) was statistically significant for IH1R1, IH1R2 and IH2R1 in trial L-1, with values $\geq$ 30% in these three treatments. The treatment with high initial water content and a moderate irrigation program showed the lowest yield loss. In trial L-2, no statistically significant losses of production were registered, with values lower than 8% in all the conditions under evaluation.

*3.3. Evaluation of the Cobweb Disease Incidence*

The impact of initial water content in the casing and irrigation program on the cobweb disease incidence was evaluated in artificially inoculated crop trials. The initial water content of the casing layers was 0.60 and 0.71 $m^3 m^{-3}$ for H1 and H2 treatment, respectively, in trial C-1, and 0.55 and 0.71 for H1 and H2 treatment, respectively, in trial C-2 (Table 1). On day 24, after the harvesting of the first flush, the irrigation program distinguished the four treatments (Figure 1d,e for Trials C-1 and C-2, respectively).

Regarding the incidence of cobweb disease in both trials, Figure 3a shows, at two different periods of the growing cycle (after harvesting of the second and third flushes), the crop surface affected by the disease for the four treatments artificially infected with conidia of *C. mycophilum*. In both trials, the disease was detected for the first time during the harvest of the second flush, with high percentage of surface affected by the disease in IH1R1 and IH1R2 treatments (17 and 26 % in trial C-1, and 19 and 14% in trial C-2), while levels were lower than 7% in both IH2 treatments; the analysis of the data showed statistical differences between treatments only for trial C-1 ($F_{3.23}$ = 12.14, *p* = 0.0001 for trial C-1, and $F_{3.23}$= 2.17, *p* = 0.1235 for trial C-2). In the third flush, although noting variability in absolute terms, the average percentage of affected surface did not report statistical differences ($F_{3.23}$ = 2.02, *p* = 0.1436 for trial C-1, and $F_{3.23}$= 2.14, *p* = 0.1268 for trial C-2).

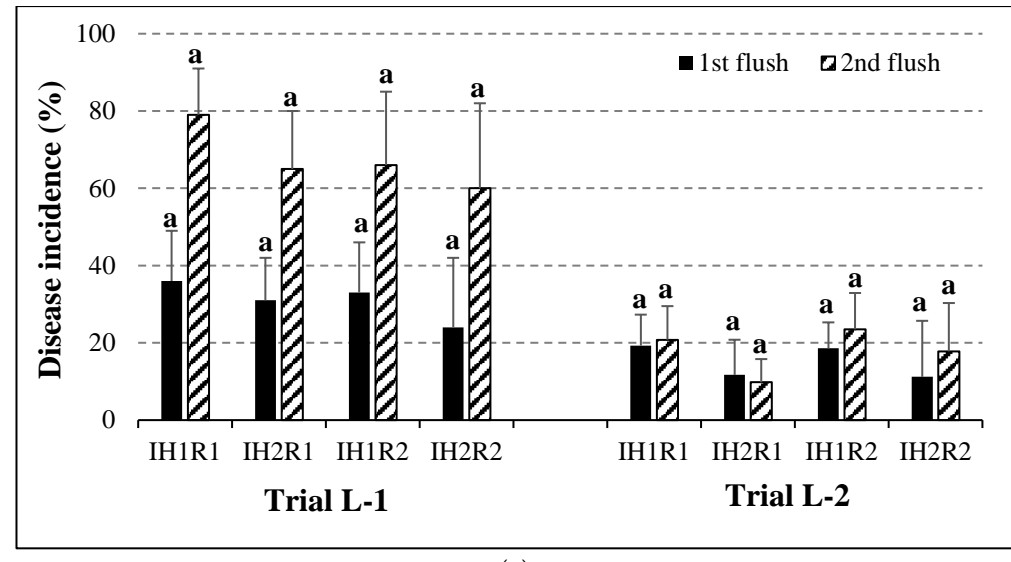

(**a**)

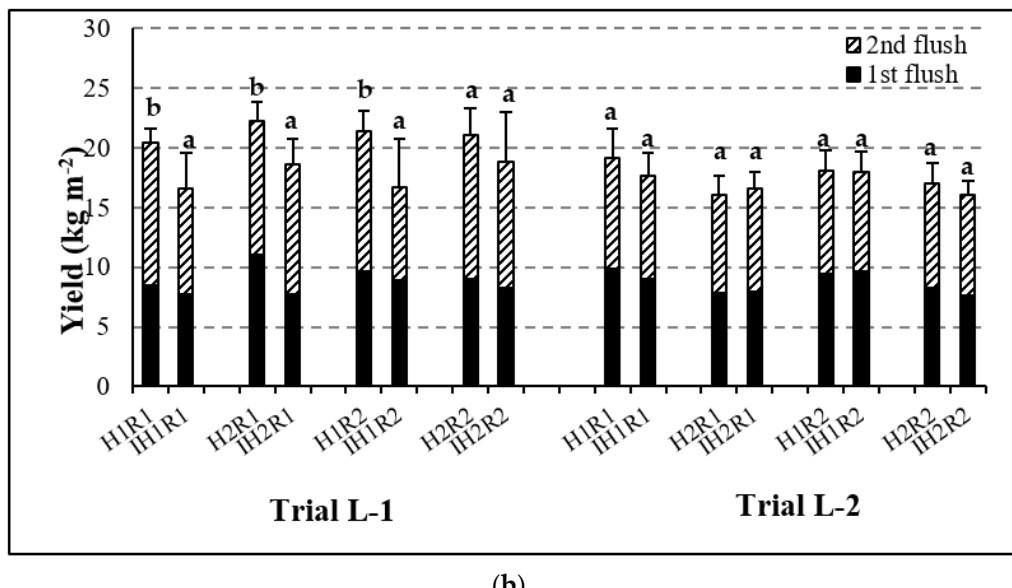

(**b**)

**Figure 2.** Pathogenic with artificial infection of *Lecanicillium fungicola* (L-1 and L-2). (**a**) Incidence; (**b**) Yield. H1R1, low initial water content with a light irrigation program; H2R1, high initial water content with a light irrigation program; H1R2, low initial water content with a moderate irrigation program; H2R2, high initial water content with a moderate irrigation program. I in treatment means artificially inoculated with the parasite. (*) For each trial and flush, the same letters show no significant differences according to Tukey's HSD test at $p = 0.05$.

Figure 3b shows the yield harvested in both trials artificially infected with cobweb disease for each treatment. Comparison between control (non-inoculated) and infected blocks (I) is plotted to evaluate the production loss due to the disease. There were statistically significant differences of cobweb-associated yield losses in the treatments with low initial water content H1R1 ($F_{1.11} = 6.96$, $p = 0.0248$ to C-1, and $F_{1.11} = 7.98$, $p = 0.0180$ to C-2) and H1R2 ($F_{1.11} = 4.90$, $p = 0.0512$ to C-1), with yield losses near to 12%. In those blocks where the casing was applied with higher initial water content, the losses registered were under 6–8%, without statistically significant differences: H2R1 ($F_{1.11} = 2.71$, $p = 0.1305$ to C-1, and $F_{1.11} = 0.10$, $p = 0.7556$ to C-2) and H2R2 ($F_{1.11} = 0.95$, $p = 0.3525$ to C-1, and $F_{1.11} = 1.13$, $p = 0.3138$ to C-2).

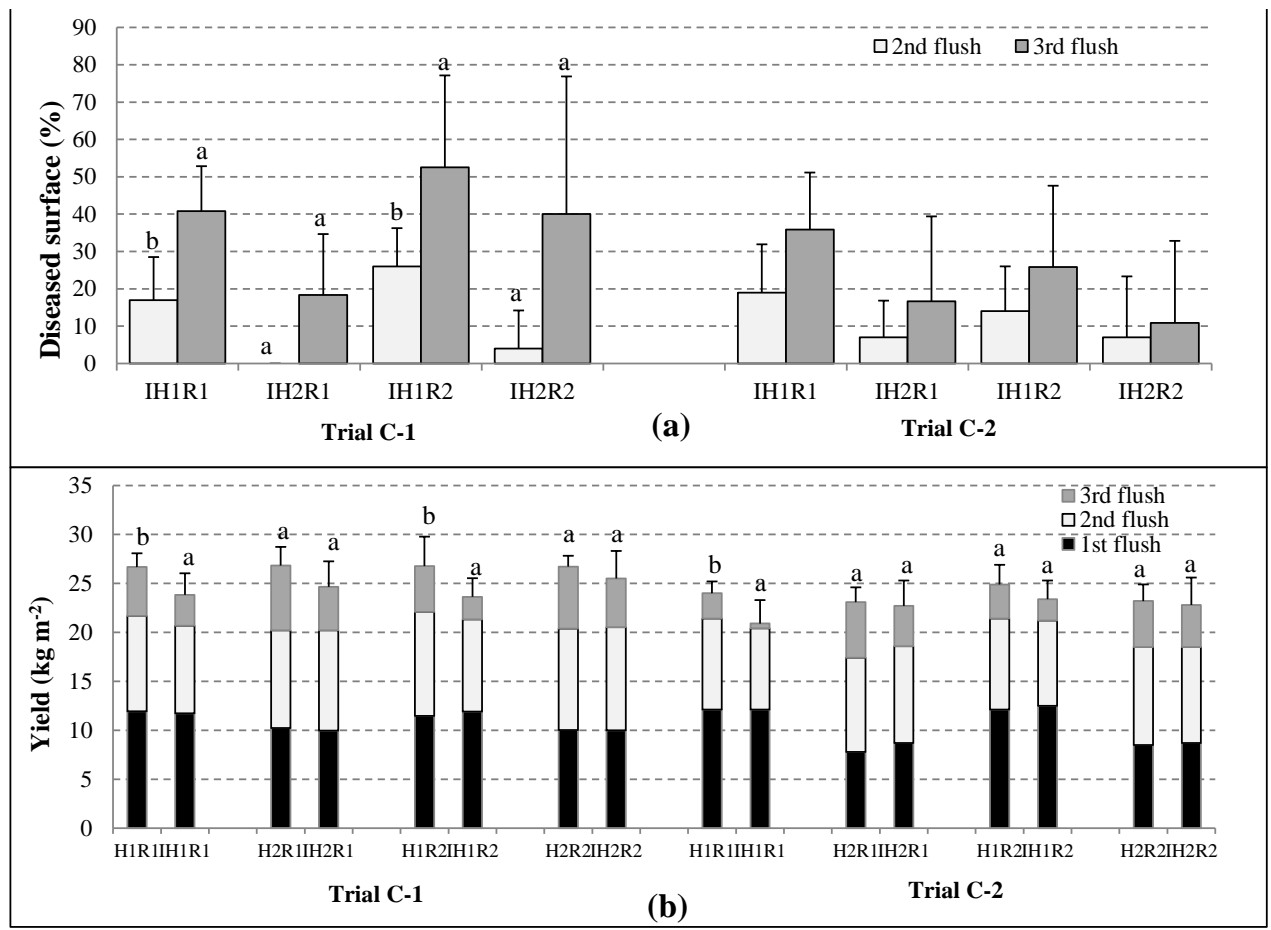

**Figure 3.** Pathogenic with artificial infection of *Cladobotryum mycophilum* (C-1 and C-2). (**a**) Incidence; (**b**) Yield. H1R1, low initial water content with a light irrigation program; H2R1, high initial water content with a light irrigation program; H1R2, low initial water content with a moderate irrigation program; H2R2, high initial water content with a moderate irrigation program. I in treatment means artificially inoculated with the parasite. (*) For each trial and flush, the same letters show no significant differences according to Tukey's HSD test at *p* = 0.05.

## 4. Discussion

The water content of the mushroom casing layer disposed on top of the colonized compost has a significant effect on mushroom productivity and quality, and a moist environment can have an impact on the germination and development of pathogenic fungi. However, this is a topic that although is highly relevant, has not been widely researched in any significant depth recently. This work aims to evaluate the effect of the initial water content in the casing and the irrigation program applied during cropping as environmental cues driving mushroom production and quality, in addition to the effect on the incidence and severity of common harmful fungal diseases, dry bubble and cobweb disease.

The amount of water held in soil or substrate can be expressed by different parameters such as the gravimetric water content (w) or volumetric water content (θ), while the availability of this material for water suction is expressed as water potential [27]. The extraction of water from the substrate and its translocation to the growing mycelia requires a decrease of total endogenous water potential. This can only be performed effectively by maintaining a water potential gradient from the substrate into the hyphal cells, which also facilitates the functioning of enzymatic machinery and nutrients uptake [30]. In this sense, *Agaricus bisporus* is able to adjust the internal solute concentrations relatively quickly to maintain the gradient and facilitate translocation of water, and enable enzymes to work effectively [19]. Therefore, the physical and chemical characteristics of the substrate determine the growth

of the mushroom mycelia and the yield of the crop, and in addition, could also determine the infection ability of the pathogens and the severity of the disease.

Dry matter content and the colour of harvested basidiomes are considered parameters for mushroom quality and are closely related to the characteristics of the casing blend used for cultivation [31]. Noble et al. reported that casing soil matric potentials producing the highest mushroom yields were related to basidiomes with lower dry matter content and tissue stiffness [16]. The negative correlation between basidiome dry matter and total mushroom yield was also noted by Barry et al., who also described an association between the higher the basidiome dry matter the poorer the colour of the basidiome [31]. Conversely, the results achieved in this paper show that the casing layer with higher level of moisture registered the highest yield, without any depreciation of dry matter and colour parameters in the carpohores. The proximity of the volumetric rate values used in the different treatments of this trial, higher at almost all times than the 58–60% (*v*/*v*) suggested by Barry et al. as the optimal casing moisture [31], could explain the values obtained for the quality parameters. Remarkably, this paper reports an important yield improvement driven by that comparatively small increase of humidity content. A higher irrigation program after the first flush also increased the production lightly, ultimately boosting the profitability of the crop.

Fletcher et al. established that one of the possible reasons for the sudden appearance of epidemic levels of cobweb disease was the use of a much finer grade of black peat with a greater WHC as a casing material [25]. The use of this product resulted in higher relative humidity and water content, and it was considered that this might have changed the environment sufficiently to encourage the development of cobweb disease. Conversely, Carrasco et al., in a survey of 2 years monitored the presence and severity of cobweb disease in commercial crops cultivated either with new peat-based casing (higher WHC) or historically employed mineral casing (lower WHC), reporting a similar incidence of cobweb disease in the mineral cased crops (34%) than in peat-based cased (29%) [12]. In the case of *C. mycophilum*, the results registered in the first trial showed that, differing from the assumption of Fletcher et al. [25], the lower the moisture in the casing layer, the more incidence of the cobweb disease, eventually resulting in higher yield loss.

The relationship between the viability of conidia and water absorbent materials was previously established for the pathogen *L. fungicola*, pointing at the peat as the primary source of this pathogen [9,32]. However, the results obtained in both trials inoculated with *L. fungicola* showed that the incidence of dry bubble disease was not related to the moisture of the casing layer at the levels evaluated. On the other hand, yield losses derived from dry bubble were more important with low initial water content and moderate irrigation.

Under conditions of higher moisture in the casing, the hydrophobic nature of fungal organisms driven by cell-wall proteins, such as hydrophobins which are present in conidia and hyphae, could delay the germination and growth of pathogenic mycelium, and therefore the occurrence of dry bubble and cobweb disease while even modifying the contact between host and parasite driven by extracellular molecules such as biosurfactants [33–35]. These results could be related not only to the moisture of the substrate, but also to the possibility of *Agaricus* mycelia being infected by the pathogenic conidia at the moment of the inoculation. In our trials, at the moment of the pathogen inoculation, those plots with an initial low moisture casing layer showed more mycelium exposed. Since the crop mycelium is hydrophobic [36], water excess delayed the initial colonization of the casing but no effect was noted in respect to the crop earliness. Of note, casing shows fungistasis against mycoparasites, but this is annulled when the host mycelium colonized the material which also deeply modified the microbiome structure of the casing [6,9]. Thus, a prompt colonisation of the casing could be related to a faster break of the fungistatic equilibrium in the casing.

Moreover, it is generally accepted that the *A. bisporus* vegetative mycelium is resistant to infection by *L. fungicola* [32]. The integrity of *A. bisporus* vegetative hyphae is not affected by *L. fungicola*, as observed in dual culture on agar medium [37,38]. However, this pathogen

is the causative agent of pin-stage abortion of *Agaricus* carphophores [39]. Largetau et al. established a significant correlation between the time needed by *A. bisporus* strains to form their first fruiting bodies and the susceptibility to *L. fungicola* [40]. Since the conidia of *L. fungicola* can survive and germinate in water in the absence of nutrients [9], regardless of the possibility of the parasite infecting the mycelium of *Agaricus* at the moment of inoculation, conidia of *L. fungicola* can remind in the casing layer until the fructification stage, when the germinative host mycelium could be infected by them. The high level of moisture of the substrate allowed the viability of the conidia in any case in our trials.

The formulation of the casing material was postulated to be a factor conditioning susceptibility to fungal diseases in mushroom crops [12]. Several bacterial strains, and natural inhabitants of the casing, have been found to show a selective suppressive effect over mycoparasites, but only minimizing disease symptoms at low pressure of disease [41]. However, after comparing three different casing materials for their natural suppressive response against dry and web bubble diseases, Carrasco et al. established that the casing material used had no significant impact on the yield obtained or the disease symptoms observed following infection with either *L. fungicola* (with heavy losses) or *Mycogone perniciosa* [6]. The results of the five trials described in this paper suggest that a right management of the moisture level of the casing materials could improve the yield of the crop and restrict the incidence of cobweb disease. In this sense, installing moisture sensors to measure the matric potential (water suction pressure) in the casing independently of the WHC of the material within the commercial growing rooms could contribute both to optimise production and prevent extensive fungal disease outbreaks [42]. As a concluding remark, the potential of keeping higher water content in the casing as a preventive treatment to cope with fungal diseases should be further evaluated for the integrated management of cobweb disease.

**Supplementary Materials:** The following are available online at https://www.mdpi.com/article/10.3390/agronomy11102063/s1, Table S1: Physical, chemical, and biological characteristics of composts used in the trials, Table S2: Physical, chemical, and biological characteristics of casing materials used in the trials.

**Author Contributions:** Conceptualization, M.J.N. and F.J.G.; methodology, M.J.N. and F.J.G.; formal analysis, M.J.N.; writing original-draft preparation, M.J.N. and F.J.G.; writing—reviewing and editing, M.J.N., J.C. and F.J.G.; visualization, J.C. and F.J.G. All authors have read and agreed to the published version of the manuscript.

**Funding:** This research was funded by MAPA (Spain) and INIA (Spain), grant numbers: RTA04-091 and RTA2010-00011-C02-01.

**Institutional Review Board Statement:** Not applicable.

**Informed Consent Statement:** Not applicable.

**Data Availability Statement:** Not applicable.

**Conflicts of Interest:** The authors declare no conflict of interest.

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
