# Peer review of "The Role of Water Content in the Casing Layer for Mushroom Crop Production and the Occurrence of Fungal Diseases"

_agronomy, doi:10.3390/agronomy11102063_

Round 1
Reviewer 1 Report
The reviewed work concerns the role of water content in the casing layer for mushroom crop production and the occurrence of fungal diseases. The work brings a lot of new and shows how to increase the yield and to some extent reduce the infection caused by pathogenic fungi. The weakest point of the manuscript is the methodical part. The methodology is poorly described and in a very incomprehensible way.
Table 1 - The table contains 5 trials and in the text above and in reference to the table there is only trial A. The description of the others is only on line 145. It is difficult to find all the necessary information.
line 123 - is described with what equipment the analysis was done, but not how it was done.
There is no control in the methodical part. Did trial A act as a control? Another control appears in the results for trials L and C, but it is not mentioned in the methodical part. It is not known whether the conditions for conducting these experiments were identical.
Why were individual trials conducted at different times of the year? Did this not affect the development of the pathogen?
When was trial A performed?
Figure 1b - part of the legend is missing
Table 2 - not all symbols are explained
line 217-219 - why the initial water content was different for trials L1 and L2 (C1 and C2 also )
Figure 2b - illegible legend why 2 bars for each combination. This is not explained in the text. There are control combinations that have not been described anywhere before.
Materials and methods - for complete improvement
Author Response
We really appreciate your comments. The methodology part has been improved, adding reference to Table 1 in the infected crop trials L1, L2, C1 and C2.
Trial A act as baseline for the agronomic performance of the crop under the different factors evaluated in the casing layer. The following statement has been added: “Trial A has been set up as a baseline to evaluate the agronomic performance of the crop.”
There is no control in the methodical part. Did trial A act as a control? Another control appears in the results for trials L and C, but it is not mentioned in the methodical part. It is not known whether the conditions for conducting these experiments were identical.
There are no-inoculated controls as stated in the M&M section: “Twelve or ten days after casing (L and C trials, respectively), 24 blocks were inoculated with a conidial suspension (7 x 103 conidia mL-1) of L. fungicola strains LF18-1 (trial L-1) and LF18-3 (trial L-2) of, or the C. mycophilum strains LR16.3 (trial C-1) and MJ-1 (trial C-2) of. Each block was sprayed (20 mL per block) by pipetting onto the surface of the casing layer at a rate of 106 conidia m-2. The remaining blocks (24 blocks) were sprayed with sterile distilled water as a control.”
Why were individual trials conducted at different times of the year? Did this not affect the development of the pathogen?
It does not affect since mushrooms are cultivated indoors under controlled environmental conditions.
When was trial A performed?
By mid-2018, included in M&M, previously to the infected trials.
Figure 1b - part of the legend is missing
Figure 1 has been corrected.
Table 2 - not all symbols are explained
The symbols have been described.
line 217-219 - why the initial water content was different for trials L1 and L2 (C1 and C2 also )
This is one of the factors under evaluation, different initial water content in the casing.
Figure 2b - illegible legend why 2 bars for each combination. This is not explained in the text. There are control combinations that have not been described anywhere before.
Controls have been described in the methodology. Bars correspond to non-infected and I=infected blocks, as described in the figure legend.
Materials and methods - for complete improvement
This section has been improved.
Reviewer 2 Report
The role of water content in the casing layer for mushroom crop production and the occurrence of fungal diseases
Navarro MJ, Carrasco J, Gea-Alegría FJ
GENERAL COMMENTS
The authors have addressed the subject of the water content of the mushroom casing layer and how that influences crop quality and yield and susceptibility to two significant button mushroom diseases. This is a topic that although is highly significant to productivity, has not been widely researched recently in any significant depth recently.
The science has been well planned and executed. But the impact has been somewhat lost in conveying results and the discussion of the results using, at times, long and convoluted sentences. I urge the authors to keep statements brief, to the point and delete any words that do not carry any weight. I have offered some suggestions in specific comments.
I congratulate the authors on addressing a significant problem in mushroom cultivation and hope they continue to investigate other moisture imbalances and quality issues.
SPECIFIC COMMENTS
- Introduction
L33 “…the selective substrate passes through…”
L36 “…of the host…”
L44 “…growers currently apply…”
L45 “…concerns advise against…”
L48 “…acts as a water…”
L50 “…the precocity of…” I suggest replacing ‘precocity’ with ‘maturity’
L52 “…important to note that an appropriate…”
L54-55 Consistency between ‘carpophore’ and ‘basidiome’
L55 “…carpophores has been…”
L56 “…of water supplied, the…”
L60 “…subsequent irrigation [20]…”
L62 “…[12], all of which also result in a reduction in…”
L65 “…was related to the…”
L67 “…ultimately creating an…” or “…ultimately providing an…”
L69 Do the authors refer to the water content of the casing or the basidiome/carpophore? Please clarify
L71-72 “…usually expressed as…”
L75 “…directly influenced by…”
L76 “…and the evidence reported indicates that the Agaricus crop…”
L79 Not sure of the meaning of ‘efficient measurements’ – rephrase
L80-81 “…dependence on chemical…”
L82 “…is a key…”
L84 “…diseases in mushroom farms…”
L84-85 “…dry bubble and cobweb.”
L85-86 Not sure what the authors are trying to say here. Perhaps “Moreover, agronomic evaluation of crop performance under these pinning conditions was also carried out to maximise crop yield.” – rephrase
- Materials and methods
L90 What was the function of Trial A? Control? Baseline comparison? Reference crop/yield?
L93 Confirm this is the same spawn strain as detailed in L138-139. Suggest replacing ‘mycelium’ with ‘strain’
L98 “…initial moisture content.”
Table 1 What was the determinant of ‘low initial water content’ versus ‘initial high water content’? for example H1 = 0.69 m3m-3 while H2 = 0.71 m3m-3. Was the water content measured through the trial at single points or a mean of several points? How do H1 and H2 compare with casing in a commercial operation?
L116 What were the parameters used to define the ‘commercial development stage’. Doesn’t this depend on whether cups or flats are defined?
L116-117 “…stage and the sporophores picked from each block were counted and weighed.”
L117 Is it possible to be consistent with the term used to refer to the mushroom? Here, It is ‘sporophore’. Refer to comment L69
L124 “…15 measurements per…”
L129 “…described in the previous section…”
L131 “…were the same as described…”
L134-135 Contract to L. fungicola and C. mycophilum as both in full previously in L41-42
L138 “…commercial strain Triple X…”
L140 “…was cased with a 35-40mm layer of a peat-based casing…”
L142 “…were placed on shelves…”
L144 Were the conidia suspended in water, PBS or other type of diluent?
L147 Clarify the method of inoculation: pipette or spray? Very different dispersal patterns
L150 Fruiting bodies – the fourth different label for the mushrooms!
L153 “…was expressed as…”
- Results
L167 “…crop (Trial A) have been evaluated.” I suggest writing ‘Trial A’ with a capital as it is the name of an important event. Throughout the text it is variously written with a lower case and an upper case ’T’. Consistency
L169 “…treatments applied in the study…”
L169 The Figure number appears to be missing from my copy of the manuscript. Please Check
Figure 1 Why did L1 trial consist of H1R1 and H2R1 only? My apologies if I have missed the explanation
L184 “…greatest moisture level…”
L185 ‘’’higher, compared to…”
L186 “…important to emphasise that...” “…initial moisture content of the casing…”
L188 “…just slightly increased the…”
L189 “…the greatest percentage…”
L191 “…flush showed no statistical difference between…”
L195 “…showing a statistically significant difference between (among?) the treatments…”
L222 “…30% in the first flush for all treatments…”
L223 “…with no statistical difference between treatments…
L224 “…but no statistical difference was detected between treatments.”
L225-226 “…there was no statistical difference between treatments, although disease incidence was lower in absolute…”
Figure 2 Pathogenicity of L. fungicola artificial infection (L-1 and L-2). X-axis legend is impossible to read on my copy of the manuscript. Please check
I can see no explanation for adopting the ‘I’ in IH1R1 for example, in materials and methods. An explanation appears first time in figure captions. My apologies if I have missed it
L234 “…letters indicate no significant difference according…”
L236 “…associated with dry…”
L239 “…no statistically significant losses…”
L242-243 “…on the incidence of cobweb disease has been…”
L246-247 I don’t understand what the authors are trying to point out here: ‘distinguished’?
L257 “…did not yield any significant difference…”
Figure 3 “Pathogenicity of C. mycophilum artificial infection (C-1 and C-2).”
L264 “…letters indicate no significant difference according…”
L271 “…losses approaching 12%...”
L272 “…where the casing with higher initial water content was applied, the losses…”
L273 …6-8% with no statistically significant difference:…”
L277 “This work evaluates the effect…” “…content of the casing…”
L279 “…of the common harmful…”
L280 “…dry bubble and cobweb.”
L282-283 “…while its availability for water suction is expressed…”
L288 “…sense, A. bisporus is…”
L291 “…determine growth of the…”
L296-300 The use of the term ‘negative correlation’ is a little confusing – perhaps rephrase these sentences. The authors’ statement is correct, but the switch from high yield and low dry matter by Noble to high dry matter and poor colour by Barry confuses two ideas in the reader’s mind.
L302 ‘depreciation’?
L303-306 Rephrase – not sure of the meaning of this statement
L306 Is ‘this paper’ referring to Barry et al or the authors’ work?
L294-309 This is a significant section of the discussion but is not easy to follow. Please rewrite this paragraph clearly and concisely
L310 “Fletcher et al proposed…”
L315 “…in a 2-year survey…”
L317 “…WHC). They reported a similar…” “…(34%) to that in peat-based…”
L321 “…fungicola, implicating peat…” or “…indicting peat…”
L330-335 Long sentence – please break this up
L336 “…mycelia being infected…”
L337-338 Could not understand this sentence – what is meant by ‘mycelium exposed’?
L340 “…but no effect…”
L340-341 “interestingly, casing demonstrates fungistasis…” “…this is nullified when…”
L343 “A prompt colonisation of the casing could therefore be…”
L344 “…break of fungistasis in the casing.”
L345 “It is generally accepted that…”
L351 ‘absence’
L352 “[9], irrespective of the…”
L353 ‘remain’
L354 ‘germinative’?
L354-356 This sentence does not make sense – rephrase
L357-358 “…a factor influencing the susceptibility of mushroom crops to fungal diseases [12].”
L362 “…against both dry and wet bubble…”
L365 Clearly differentiate ‘this paper’ from the work being reported herein or Carrasco et al
L365 “…suggests that the correct management of casing moisture levels…”
L371 “more studies are necessary…” “…standardize cultural practices to…”
Author Response
Reviewer 2
We really appreciate the comments of the reviewer. Moderate English changes required have been checked. Please find below the answer to some comments.
L50 “…the precocity of…” I suggest replacing ‘precocity’ with ‘maturity’
It has been replace with “earliness” because it is the term usually applied for mushroom harvest. It does not refers to the maturity of the sporophores, but to the time between the application of the casing and the harvesting of the first mushrooms (therefore earliness of the crop). Higher earliness means that the farmers will harvest promptly and ultimately they will icrease the number of crop cycles during the year.
L54-55 Consistency between ‘carpophore’ and ‘basidiome’
“Carpohore, sporophore, basidiome, mushroom”, those terms are synonyms. In order to facilitate the reading the term basidiome has been employed instead of carpophore or sporophore.
L69 Do the authors refer to the water content of the casing or the basidiome/carpophore? Please clarify
It refers to water content in the casing layer. It has been clarified
L90 What was the function of Trial A? Control? Baseline comparison? Reference crop/yield?
Trial A was carried out as a baseline to test the agronomic valuation of the crop performance under these different casing moisture conditions. The following sentence has been included in M&M: Trial A has been set up as a baseline to evaluate the agronomic performance of the crop.
L93 Confirm this is the same spawn strain as detailed in L138-139.
They were not the same spawn. Trial A was carried out growing “laboratorio” spawn (Champinter…..), and Trials L-1, L-2, C-1 and C-2 used Triple X spawn (Amycel….).
Table 1 What was the determinant of ‘low initial water content’ versus ‘initial high water content’? for example H1 = 0.69 m3m-3 while H2 = 0.71 m3m-3.
H1 was the water content of the casing material just got from the supplier. H2 was the water content value after our moisturizing, therefore supplying a higher level.
Was the water content measured through the trial at single points or a mean of several points?
The volumetric water content was monitored through the crop cycle (including the initial value) using a GP1 Datalogger with SM-150T sensor, only one (fixed) sensor by treatment.
How do H1 and H2 compare with casing in a commercial operation?
The water content within the casing layer must be monitored.
L117, L150 Is it possible to be consistent with the term used to refer to the mushroom? Here, It is ‘sporophore’. Refer to comment L69
The term “basidiome” has been used instead mushroom, sporophore or carpophore when referring to the fruiting bodies. The term mushroom has been kept to referring to the agronomy of the crop, for instance: “mushroom productivity”; “mushroom crop”, “mushroom mycelium”.
L169 The Figure number appears to be missing from my copy of the manuscript. Please Check
- This sentence has been removed; it was duplicated
Figure 1 Why did L1 trial consist of H1R1 and H2R1 only? My apologies if I have missed the explanation –
Sorry, it was an edition mistake. It has been improved.
I can see no explanation for adopting the ‘I’ in IH1R1 for example, in materials and methods. An explanation appears first time in figure captions. My apologies if I have missed it
You are right. It has been explained in materials and methods section
L246-247 I don’t understand what the authors are trying to point out here: ‘distinguished’?
It has been replaced by “differentiated”
L294-309 This is a significant section of the discussion but is not easy to follow. Please rewrite this paragraph clearly and concisely (L296-300 The use of the term ‘negative correlation’ is a little confusing – perhaps rephrase these sentences. The authors’ statement is correct, but the switch from high yield and low dry matter by Noble to high dry matter and poor colour by Barry confuses two ideas in the reader’s mind)
The parragraph has been rewritten
L330-335 Long sentence – please break this up
The parragraph has been rewritten
L354-356 This sentence does not make sense – rephrase
The sentence has been rephased
L365 Clearly differentiate ‘this paper’ from…..
The sentence has been improved.
Reviewer 3 Report
The manuscript submitted for review should be corrected and supplemented. The abstract needs to be redrafted, it is currently incomprehensible. We do not number keywords. Correctly written introduction, however, is missing and what the research was for - a clearly formulated research goal. There is no information in the methodology whether the substrate was sterilized, and if so, how? The culture was infected with the pathogen how? Was it reisolated at the end? Unfortunately, without this data, it is difficult for me to judge the correctness of the results. As a result, the title does not correspond to the content of the work. The manuscript should be completed.
Author Response
Reviewer 3:
We really appreciate the comments of the reviewer, please find below the reponse to some particular comments.
Abstract- The abstract needs to be redrafted, it is currently incomprehensible.
We are sorry to hear this appreciation but our opinion is that the abstract is understandable as far as you understand the nature of the crop, which means that it is required to understand the casing as a substrate for mushroom cutivation. As described in the first section of the introduction: “The commercial cultivation of button mushroom (Agaricus bisporus (Lange) Imbach) is an intensive horticultural process that requires two different substrates: a selective compost, which is a nutritive reservoir for mycelium growth and development, and the casing layer, that is placed on top of the colonized compost and built up with specific materials to induce and favor the development of basidiomes”. In addition, to specify the role of the casing as water reservoir we have added the following statement in the Introduction section: “A correct water content in the casing is required to provide the essential moist microclimate that supplies water for the growth and development of the crop while facilitating the transport of dissolved nutrients and prevents compost surface from drying out [1-2].”
Keywords- We do not number keywords
Keyword numbers had been removed.
Introduction- Correctly written introduction, however, is missing and what the research was for - a clearly formulated research goal
By the end of the introduction section we summarise the research purpose of the article. In addition, following the reviewer recommendations, we have added a paragraph to specify the impact of the results achieved on mushroom production and fungal disease control:
This paper analyses the effect of different levels of moisture in the casing layer in relation to the incidence of two of the most important fungal diseases in the mushroom growing farms, dry bubble and cobweb disease. Moreover, agronomic valuation of crop performance under these setting conditions was also carried out to maximise crop yield. Ultimately, the results achieved contribute to understand the role of the water content in the casing to maximise mushroom yield and as a cultural input to prevent fungal infections.
Methodology- There is no information in the methodology whether the substrate was sterilized, and if so, how?
We have used ready to case phase III compost. This is high quality compost coming from the composting maker, free of diseases and competitors and fully colonized by the mycelium as boadly explained in the literature (i.e. Fletcher, J.T.; Gaze, R.H. Mushroom Pest and Disease Control. Manson Publishing. London, UK, 2008. 192 pp.). Therefore, the compost was not sterilised.
The culture was infected with the pathogen how? Was it re-isolated at the end?
The inoculation process with the different parasites is described in the Materials and methods section, please double check:
“Twelve or ten days after casing (L and C trials, respectively), 24 blocks were inoculated with a conidial suspension (7 x 103 conidia mL-1) of L. fungicola strains LF18-1 (trial L-1) and LF18-3 (trial L-2) of, or the C. mycophilum strains LR16.3 (trial C-1) and MJ-1 (trial C-2) of. Each block was inoculated (20 mL per block) by pipetting onto the surface of the casing layer at a rate of 106 conidia m-2.”
Reisolation of parasites was not required since the symptoms of disease registered are fully associated to the different causative agents inoculated as described (“In the case of L. fungicola trials, the mushrooms and dry bubbles were separately harvested daily during two flushes. Harvested mushrooms were expressed as either healthy or infected by L. fungicola. Disease incidence was stated as the ratio of diseased sporophores vs the total number of harvested mushrooms (healthy and diseased) [28]. The effect of cobweb on mushroom productivity was evaluated by comparing mush-room yield (kg m−2), harvested during three flushes, and also based on cobweb patches detected on the casing material to quantify the area colonized by cobweb as previously detailed [12, 29]”)
We have extensively used this same approach in previous research published in Q1 journals.
Round 2
Reviewer 3 Report
thank you for replying to the review.